# External Validation of Deep Learning Models for Classifying Etiology of Retinal Hemorrhage Using Diverse Fundus Photography Datasets

**DOI:** 10.3390/bioengineering12010020

**Published:** 2024-12-29

**Authors:** Pooya Khosravi, Nolan A. Huck, Kourosh Shahraki, Elina Ghafari, Reza Azimi, So Young Kim, Eric Crouch, Xiaohui Xie, Donny W. Suh

**Affiliations:** 1Department of Ophthalmology, School of Medicine, University of California, Irvine, CA 92697, USA; pooyak@hs.uci.edu (P.K.); nahuck@hs.uci.edu (N.A.H.); kourosh.shahyar@gmail.com (K.S.); eghafari@hs.uci.edu (E.G.); razimi1@hs.uci.edu (R.A.); 2Donald Bren School of Information and Computer Sciences, University of California, Irvine, CA 92697, USA; xhx@uci.edu; 3Department of Ophthalmology, Soonchunhyang University College of Medicine, Cheonan, 31151, Republic of Korea; ophdrkim@gmail.com; 4Department of Ophthalmology, Eastern Virginia Medical School, Norfolk, VA 23507, USA; ercrouch@gmail.com

**Keywords:** retinal hemorrhage (RH), deep learning (DL), fundus photography, external validation

## Abstract

Retinal hemorrhage (RH) is a significant clinical finding with various etiologies, necessitating accurate classification for effective management. This study aims to externally validate deep learning (DL) models, specifically FastVit_SA12 and ResNet18, for distinguishing between traumatic and medical causes of RH using diverse fundus photography datasets. A comprehensive dataset was compiled, including private collections from South Korea and Virginia, alongside publicly available datasets such as RFMiD, BRSET, and DeepEyeNet. The models were evaluated on a total of 2661 images, achieving high performance metrics. FastVit_SA12 demonstrated an overall accuracy of 96.99%, with a precision of 0.9935 and recall of 0.9723 for medical cases, while ResNet18 achieved a 94.66% accuracy with a precision of 0.9893. A Grad-CAM analysis revealed that ResNet18 emphasized global vascular patterns, such as arcuate vessels, while FastVit_SA12 focused on clinically relevant areas, including the optic disk and hemorrhagic regions. Medical cases showed localized activations, whereas trauma-related images displayed diffuse patterns across the fundus. Both models exhibited strong sensitivity and specificity, indicating their potential utility in clinical settings for accurate RH diagnosis. This study underscores the importance of external validation in enhancing the reliability and applicability of AI models in ophthalmology, paving the way for improved patient care and outcomes.

## 1. Introduction

Retinal hemorrhage (RH) is a critical clinical finding that can arise from a variety of etiologies, ranging from traumatic causes such as abusive head trauma (AHT) in children to medical conditions like diabetic retinopathy (DR), age-related macular degeneration (AMD), and other retinal vascular diseases [1]. The differentiation between traumatic and medical causes of RH is not only pivotal for appropriate clinical management but also for legal consideration and social interventions [1,2].

In recent years, the field of deep learning (DL) in medical imaging has shown promise in the automated classification of retinal conditions, offering potential tools for aiding in the diagnosis and understanding of RH. However, the generalizability and external validity of these models across diverse fundus photography datasets remain a significant concern [1].

A systematic review by Kim et al. [3] revealed that only 6% of publications on AI in medical imaging incorporated external validation, while Yao et al. [4] and Nguyen et al. [5] reported similar findings in their respective reviews of deep learning applications in neuroradiology and machine learning algorithms for differentiating glioblastoma multiforme from primary central nervous system lymphoma. These statistics underscore the critical need for comprehensive validation to ensure the reliability and applicability of AI models in clinical settings.

The burgeoning field of DL in medical imaging has led to the development of various models capable of detecting and classifying retinal diseases with high levels of accuracy. For instance, studies have demonstrated the efficacy of DL models in identifying different stages of DR [6], detecting retinal hemorrhage in head computed tomography (CT) scans [7], and classifying retinal nerve fiber and choroid layers as markers of AMD [8]. Moreover, the generation of structurally realistic retinal fundus images using diffusion models [9] and the classification of intracranial hemorrhage from CT scans [10] highlight the versatility of DL applications in ophthalmology and neurology.

Despite these advancements, the challenge of validating these DL models on external datasets persists. The performance of DL models can be significantly affected by the heterogeneity of data sources, variations in image quality, and differences in disease prevalence across datasets. For example, the development of a deep learning model for retinal hemorrhage detection on pediatric head CT images [7] underscores the need for external validation to ensure the model’s reliability and prevent overfitting. Similarly, the classification of retinal vascular diseases using ultra-wide field color fundus photographs [11] and the human-in-the-loop retinal image classification with a customized loss function [12] emphasize the importance of robust validation techniques to ensure the models’ applicability in real-world settings. Consequently, to evaluate the real-world clinical efficacy, it is imperative to understand an algorithm’s performance on an external dataset that originates from a distinct source, separate from the development data and not utilized in the algorithm’s training. Although the significance of external validation in artificial intelligence research is gaining recognition, it has been implemented in a limited number of published studies [13].

In addition to external validation, the need for diverse data representation is increasingly evident. For example, the Brazilian Multilabel Ophthalmological Dataset (BRSET) addresses the scarcity of publicly available datasets in underrepresented regions by providing over 16,000 color fundus images and sociodemographic information, enabling performance analysis across diverse groups [14]. Such datasets exemplify the importance of incorporating data from varied sources to enhance model robustness and ensure reliable generalizability in clinical applications. These efforts underscore the critical role of diversity and external validation in advancing the development and deployment of DL models in real-world settings.

In light of these considerations, this paper aims to conduct an external validation of DL models for classifying the etiology of retinal hemorrhage into traumatic or medical causes using diverse fundus photography datasets. By leveraging datasets from various sources, including those used for detecting and classifying DR [6,15,16], AMD [8,12,17], and other retinal vascular diseases [11], we seek to evaluate the models’ performance and generalizability. This external validation is crucial for assessing the models’ diagnostic utility and for ensuring that they can be reliably deployed in clinical practice to assist healthcare professionals in making informed decisions regarding the management of patients with RH.

## 2. Methods

### 2.1. Data Collection

For the external validation of our machine learning model, which classifies the etiology of retinal hemorrhages in fundus photographs into trauma or medical etiology, we utilized a combination of private and public datasets.

#### Private Dataset Collection

Image labels were assigned based on the underlying case diagnosis. The determination of hemorrhage presence or absence was entrusted to a panel of ophthalmologists (E.G., R.A., S.Y.K., E.C., and D.W.S.). In cases of differing interpretations of an image, consensus was reached through careful discussion and deliberation to arrive at the final classification. The private dataset included cases of confirmed AHT as determined by a multidisciplinary child abuse team. Additionally, it comprised medically validated cases correlated with laboratory results, and accidental trauma cases confirmed by witness accounts, scene investigations, and consistent physical findings. Cases with uncertain etiology were excluded. Since all images were unrecognizable and anonymized, Institutional Review Board (IRB) approval was not required. Data-sharing and access comply with the respective regulations of the source institutions, and data usage was limited strictly to research purposes as agreed upon with the data providers. The photographs were sourced from the personal collections of the authors.

### 2.2. Data Preprocessing

To ensure uniformity and enhance variability across images from different sources and devices, all images were resized to 256 × 256 pixels using bilinear interpolation, with zero-padding applied as needed. This resizing step provided a consistent resolution for data augmentation during training, which included random contrast adjustment, Gaussian noise, horizontal flipping, cropping, and cutout techniques to improve model robustness. Following augmentation, images were resized to 224 × 224 pixels to match the input size requirements of the transfer learning framework used in our study. The normalization per channel, based on the ImageNet training set, was applied at all stages. During testing, images were resized directly to 224 × 224 pixels and underwent the same normalization to ensure consistency with the model’s input size and training pipeline.

### 2.3. Model Training

The training of our deep learning models, aimed at distinguishing between retinal hemorrhages of traumatic and medical origins, utilized convolutional neural networks (CNNs) and Transformer architectures. The models included ResNet-based architectures (ResNet18) [18] and a hybrid vision transformer architecture (FastViT) [19]. Each model was fine-tuned after initialization with pretrained ImageNet weights from the PyTorch repository [20]. The model was then trained on 597 images, comprising 298 medical cases (49.9%) and 299 trauma cases (50.1%). Full details on the model architectures and training procedures, are available in our previous study [1].

### 2.4. Evaluation

The models were evaluated on this external dataset using established performance metrics such as sensitivity, specificity, and overall accuracy. The area under the receiver operating characteristic curve (AUC) was also calculated to assess the models’ ability to distinguish between traumatic and medical retinal hemorrhages. The Youden Index threshold, derived from the original model, was used to calculate the sensitivity, specificity, positive predictive value (PPV), and negative predictive value (NPV). This threshold was chosen for its ability to maximize the difference between the true positive rate (sensitivity) and the false positive rate (1-specificity), thereby facilitating a comprehensive assessment of the models’ performance.

### 2.5. Interpretability Using Grad-CAM

To enhance the interpretability of the deep learning models, gradient-weighted class activation mapping (Grad-CAM) [21] was utilized to visualize the regions of fundus images that contributed most to the predictions made by the models. Grad-CAM generates heatmaps by identifying and highlighting spatial areas that have the greatest influence on the classification outcome. This method was applied to the final convolutional layers of both ResNet18 and FastViT-SA12 models. This systematic approach enabled a deeper understanding of the models’ decision-making processes and their ability to capture relevant diagnostic features.

## 3. Results

### 3.1. Datasets

To ensure clinical relevance and enhance the diversity and robustness of our validation process, we incorporated three publicly available datasets and collected private datasets from two distinct regions. Datasets are summarized in Table 1. Figure 1 shows examples from the combined validation dataset.

#### 3.1.1. Private Datasets

These datasets, curated with comprehensive clinical contexts, were anonymized to protect patient confidentiality. The South Korean Dataset includes 114 images of retinal hemorrhages from spontaneous vaginal delivery, representing non-pathological trauma-related cases. Collected from various medical institutions in South Korea, these images broaden the representation of trauma for model validation. Future studies could include instrumental deliveries to enhance the diversity and generalizability of the trauma class. The Virginia Dataset consists of 192 images of retinal hemorrhages caused by abusive head trauma (AHT), sourced from medical facilities in Virginia, USA. Like the South Korean dataset, all images were anonymized to ensure patient privacy.

#### 3.1.2. Public Datasets

The public datasets mostly contained retinal hemorrhages due to a medical condition and contained very few examples of traumatic retinal hemorrhage. For each dataset, images, with labels and notes if available, were reviewed and those with retinal hemorrhage were selected. The following are the public datasets:

RFMiD + RFMiD 2.0 Dataset [22,23]: The Retinal Fundus MultiDisease Image Dataset contains images of various retinal diseases, aiding in the generalization of our model. After review, this dataset contained 1918 RH with medical etiology (diabetic retinopathy, hemorrhagic retinopathy, central retinal vein occlusion, pre-retinal hemorrhage, branch retinal vein occlusion, retinitis, macroaneurysm, and hemorrhagic pigment epithelial detachment) and 6 RH with traumatic etiology (post-traumatic choroidal rupture).

**Table 1 bioengineering-12-00020-t001:** Summary of the datasets used in the study, categorized by etiology of retinal hemorrhage (medical vs. trauma) and source. The table includes both private datasets (South Korea and Virginia) and public datasets (DeepEyeNet [24], RFMiD [23], and BRSET [14]).

Dataset	Medical Etiology (*n* = 2346)	Trauma Etiology (*n* = 315)
South Korea Dataset (*n* = 114)	0	114
Virginia Dataset (*n* = 192)	0	192
DeepEyeNet (*n* = 335)	332	3
RFMiD (*n* = 1924)	1918	6
BRSET (*n* = 96)	96	0

BRSET Dataset [14]: The Brazilian Multilabel Ophthalmological Dataset includes images from a diverse patient population, further enhancing the external validity of our model. Images that had the tag hemorrhage (*n* = 96) were selected from the total of 16,266 images in the dataset.

DeepEyeNet Dataset [24]: The DeepEyeNet dataset is a comprehensive collection of retinal images used to train and validate the DeepEyeNet model. This dataset includes a wide range of retinal hemorrhage cases with detailed annotations, providing a rich resource for model development and external validation. The diversity of the dataset ensures that the model is exposed to various manifestations of retinal hemorrhages, thereby improving its diagnostic accuracy and generalizability. Three images were determined to be due to traumatic cause based on the noted history, and 332 images were selected with retinal hemorrhages due to a medical condition.

### 3.2. Model Performance on External Datasets

The performance of two models, FastVit_SA12 and ResNet18, was evaluated on an external dataset comprising 2661 images to distinguish the underlying etiology of retinal hemorrhages in fundus photographs, specifically between medical and trauma causes.

The FastVit_SA12 model demonstrated high performance, achieving an overall accuracy of 96.99%. For medical cases, the model exhibited a precision of 0.9935 and a recall of 0.9723, while for trauma cases, it showed a precision of 0.8219 and a recall of 0.9524. The confusion matrix (Appendix A) highlights that the model correctly identified 2281 medical and 300 trauma cases. The AUC was 0.9811 (Appendix A), and the Youden Index indicated a specificity of 97.23% and a sensitivity of 95.56%. The ResNet18 model also performed well, with an overall accuracy of 94.66%. For medical cases, the model achieved a precision of 0.9893 and a recall of 0.9497. For trauma cases, the precision was 0.7115, and the recall was 0.9238. The confusion matrix (Appendix A) shows that the model correctly identified 2228 medical and 291 trauma cases. The AUC for ResNet18 was 0.9626 (Appendix A), and the Youden Index indicated a specificity of 94.97% and a sensitivity of 92.70%. Comprehensive performance metrics are presented in Table 2.

Further analysis of the misclassifications (Figure 2) revealed key patterns. FastVit-SA12 missed eight AHT cases, one birth trauma, and six choroidal ruptures from the RFMiD test set. Similarly, the ResNet18 model missed 17 AHT cases, 1 birth trauma, and 6 choroidal ruptures. Notably, both models failed to correctly classify all trauma cases from the RFMiD dataset, suggesting the need for greater representation of trauma-related retinal hemorrhages in public datasets to improve model robustness.

The missed cases followed identifiable patterns. Misclassified images were often out of focus or exhibited poor lighting, resulting in incomplete visibility of the optic disk and arcuate vessel structures. Trauma cases with minimal retinal hemorrhages (RH), particularly around the macula, were frequently misdiagnosed as medical. Conversely, medical cases with widespread hemorrhages, such as those caused by retinal vein occlusions or severe myopia, were sometimes classified as trauma.

Overall, both models successfully distinguished the etiology of retinal hemorrhages in fundus photographs. The ROC curves for both models (Appendix A) further highlight their diagnostic capabilities, with FastViT-SA12 showing superior metrics across most evaluation parameters compared to the ResNet18 model, suggesting it may be more effective for this classification task.

### 3.3. Interpretability Analysis

The Grad-CAM visualizations revealed notable differences in the focus areas of the two models. ResNet18 primarily emphasized the arcuate vessels across the fundus (Figure 3A right), which may indicate that the model uses structural vascular patterns to distinguish between medical and trauma-related hemorrhages. In contrast, FastViT-SA12 consistently highlighted the optic disk and specific focal regions within the fundus (Figure 3A middle). This behavior suggests that FastViT-SA12 is more sensitive to localized pathological features, which are often of greater clinical relevance in identifying retinal hemorrhages. The arcuate vessel focus seen in ResNet18 may reflect the model’s learning of global structural patterns, while FastViT-SA12’s optic disk focus indicates a preference for critical regions commonly evaluated in clinical practice.

Distinct activation patterns emerged when Grad-CAM heatmaps were compared between medical and trauma-related retinal hemorrhage images. In medical cases, the highlighted areas were generally focused and localized, such as the optic disk, arcuate vessels, and specific regions of hemorrhage. Conversely, in trauma-related images, the Grad-CAM outputs demonstrated a diffuse and widespread activation across the fundus (Figure 3B), rather than being limited to specific regions. This broad activation suggests that the models detect subtle, widespread disruptions in the retinal structure, which are characteristic of traumatic hemorrhages.

An unexpected finding emerged in cases of retinal vein occlusion. While prominent hemorrhagic areas were visible in the images, the Grad-CAM heatmaps did not emphasize these regions. Instead, the models highlighted other areas, such as adjacent vascular structures or the optic disk, as shown in Figure 3C. This behavior suggests that the models may rely on secondary or global patterns rather than the hemorrhagic areas themselves for classification. These findings highlight the complexity of the decision-making process and the need for further investigation to understand how models prioritize specific features in certain conditions.

## 4. Discussion

This study aimed to externally validate two advanced deep learning models, ResNet18 and FastViT-SA12, for classifying retinal hemorrhage etiologies using the largest dataset of retinal hemorrhages due to traumatic events. The comprehensive and diverse dataset allowed for robust external validation, demonstrating that both models possess high accuracy, sensitivity, specificity, and AUC values, indicating strong performance in classifying retinal hemorrhages into traumatic or medical causes. These findings underscore the potential of advanced deep learning techniques in enhancing diagnostic processes in ophthalmology, particularly in distinguishing between different etiologies of retinal hemorrhages, which is crucial for patient management and treatment planning.

An in-depth interpretability analysis using gradient-weighted class activation mapping (Grad-CAM) revealed significant differences in the decision-making processes of ResNet18 and FastViT-SA12. ResNet18 predominantly focused on arcuate vessels, suggesting a reliance on global structural features to differentiate between medical and traumatic cases. In contrast, FastViT-SA12 highlighted clinically relevant regions, such as the optic disk and focal hemorrhagic areas, aligning more closely with diagnostic patterns observed in clinical practice [25,26,27,28,29]. These distinct behaviors reflect differences in how the models learn and prioritize visual features for classification. When the Grad-CAM outputs were analyzed for medical versus trauma-related hemorrhages, clear differences emerged. Medical cases exhibited focused activations on the optic disk, arcuate vessels, or localized hemorrhagic areas, consistent with specific pathologies such as diabetic retinopathy or retinal vein occlusion [30]. Conversely, trauma-related images demonstrated diffuse and widespread activations across the fundus, reflecting the global disruptions in retinal structure commonly seen in traumatic events [27,29]. Interestingly, in cases of retinal vein occlusion, the Grad-CAM heatmaps did not emphasize the hemorrhagic areas directly. Instead, both models highlighted secondary regions such as adjacent vascular structures or the optic disk, suggesting that the models relied on global or secondary features for classification. These findings underscore the models’ ability to capture subtle patterns differentiating etiologies while also highlighting opportunities for improvement, such as developing frameworks that prioritize primary pathological regions more effectively.

Analysis of misclassified cases revealed consistent patterns that provide insights into the models’ limitations. Both FastViT-SA12 and ResNet18 struggled with low-quality images, particularly those that were out of focus or exhibited poor lighting conditions, which obscured critical regions such as the optic disk and arcuate vessels. This issue was more prominent in trauma cases with minimal hemorrhages, such as those confined to the macula, where the subtle nature of the findings led to misclassification as medical etiology. Conversely, medical cases with widespread hemorrhages, particularly those caused by conditions like retinal vein occlusion or severe myopia, were sometimes misinterpreted as trauma due to the global spread of hemorrhagic patterns.

A closer examination revealed that FastViT-SA12 outperformed ResNet18 in challenging cases, particularly in trauma classification. FastViT-SA12 missed 8 AHT cases, 1 birth trauma, and 6 choroidal ruptures, while ResNet18 missed 17 AHT cases, 1 birth trauma, and the same 6 choroidal ruptures from the RFMiD test set. Notably, both models failed to classify all RFMiD trauma cases, suggesting that public datasets currently lack sufficient examples of trauma-related hemorrhages, particularly subtle presentations like choroidal ruptures.

These findings emphasize the importance of improving both data quality and diversity. Incorporating higher-resolution images, ensuring consistent imaging standards, and expanding datasets with trauma cases of varying severity are essential to enhance model robustness. Furthermore, misclassified examples revealed that the models sometimes relied on secondary structural features, such as adjacent vessels or the optic disk, instead of focusing on primary hemorrhagic regions. Advancing interpretability tools and refining attention mechanisms could help models prioritize the most clinically relevant regions, thereby reducing misclassifications.

Integrating these models into real-world clinical workflows has significant potential to enhance diagnostic workflows in ophthalmology. They could assist in triaging patients in emergency and outpatient settings by prioritizing high-risk cases and optimizing resource allocation. In telemedicine settings, they could support remote diagnostics, particularly for underserved populations. Additionally, embedding these models into electronic health record (EHR) systems could enable seamless, automated analysis of fundus images as part of routine workflows. However, while these models demonstrate strong accuracy and specificity, caution is necessary for medico-legal applications. Given the high stakes involved, the models should only be used as adjunctive tools alongside clinical judgment and multidisciplinary team evaluations to mitigate risks associated with potential misclassifications.

Despite its strengths, this study has limitations. The external validation datasets included diverse populations, but underrepresented groups remain, limiting the full generalizability of the models. Expanding datasets to achieve a more global representation is crucial for improving real-world applicability. As with any machine learning model, the possibility of encountering unique or rare cases not represented in the training or validation datasets remains. The variability in focus regions identified by Grad-CAM in certain cases, such as retinal vein occlusion, highlights the need for further refinement of the models to ensure they consistently prioritize primary pathological features over secondary or global patterns. The variability in performance across datasets, as highlighted in prior studies, further underscores the importance of external validation to ensure robustness. This principle is particularly applicable to models classifying the etiology of retinal hemorrhages, where integrating diverse data sources during development is critical for addressing heterogeneity in clinical presentations. Without such validation, the reliability of AI models in real-world settings cannot be guaranteed [31,32,33]. Furthermore, the misclassification of certain cases, although low, suggests areas for model improvement, particularly in differentiating complex cases that may present with overlapping characteristics of both traumatic and medical etiologies. By increasing the dataset size and incorporating more diverse patient populations, the model could be further refined to better handle challenging or ambiguous cases, thereby enhancing its robustness and diagnostic accuracy.

Ongoing integration into clinical workflows can provide valuable feedback for further refinement and optimization of these models. Testing in real-world settings across diverse healthcare environments will help identify areas for improvement, enhance generalizability, and ensure the models’ practical utility. Addressing the limitations observed, such as misclassifications in low-quality images and subtle traumatic hemorrhages, will require incorporating higher-quality imaging data and exploring the inclusion of multi-modal information such as OCT imaging, patient history, or clinical notes to provide a more comprehensive diagnostic perspective. Further, advancements in model interpretability—through tools like Grad-CAM—will be critical for fostering clinician trust, improving transparency, and identifying areas where models prioritize global or secondary features over primary pathological regions. By refining attention mechanisms and enhancing models’ ability to focus on clinically relevant areas, such as hemorrhagic zones or subtle abnormalities, AI systems can achieve greater accuracy and reliability. Through ongoing advancements in data quality, model architecture, and prospective clinical validation, these tools have the potential to significantly enhance diagnostic workflows, improve patient outcomes, and ensure broader adoption of AI in ophthalmology and healthcare.

## 5. Conclusions

The external validation of the ResNet18 and FastViT-SA12 models confirms their efficacy in classifying retinal hemorrhages with high accuracy, sensitivity, and specificity. FastViT-SA12 demonstrated superior performance, particularly in identifying trauma-related hemorrhages, underscoring its potential to enhance clinical decision making in ophthalmology. However, challenges such as low-quality imaging, subtle hemorrhages in trauma cases, and complex medical presentations highlight areas requiring further improvement.

To address these challenges, improvements in data quality are essential, including high-resolution, well-annotated datasets that capture diverse imaging conditions and subtle pathological variations. Additionally, integrating multi-modal approaches—such as combining fundus images with OCT imaging or patient history—could provide a more comprehensive representation of retinal pathologies, improving model accuracy. Advancing model architectures through attention mechanisms and region-specific learning frameworks would enable models to prioritize critical pathological features over global or secondary structures, as demonstrated in Grad-CAM analyses. Enhancing interpretability tools will further help clinicians understand model decision making, fostering trust and broader clinical adoption.

Through ongoing advancements in data quality, model architecture, and clinical validation, these AI tools can achieve greater robustness, interpretability, and reliability. Prospective, real-world studies in diverse clinical settings are crucial to ensure the seamless integration of these models into routine workflows. Ultimately, these improvements will pave the way for enhanced diagnostic accuracy, optimized workflows, and improved patient outcomes in ophthalmology and broader medical imaging applications.

## Figures and Tables

**Figure 1 bioengineering-12-00020-f001:**
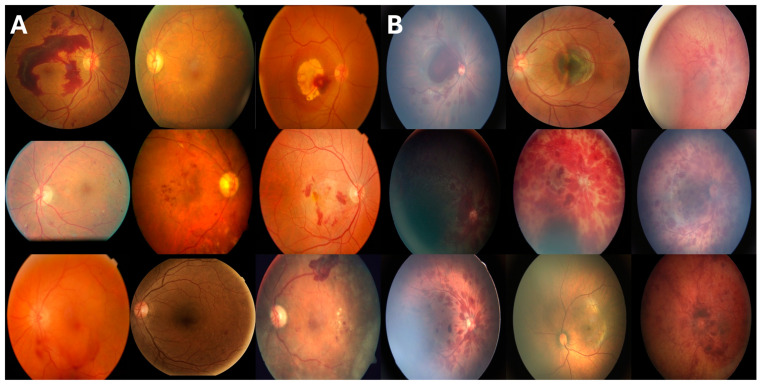
Representative fundus images from the combined external validation dataset. (**A**) The left panel shows 9 examples of retinal hemorrhages due to **medical etiologies** (retinal vein occlusions, diabetic retinopathy, and retinitis), while (**B**) the right panel displays 9 examples of retinal hemorrhages due to **trauma etiologies** (post-traumatic choroidal rupture, abusive head trauma, and birth trauma). The images demonstrate the range of variations observed across both categories.

**Figure 2 bioengineering-12-00020-f002:**
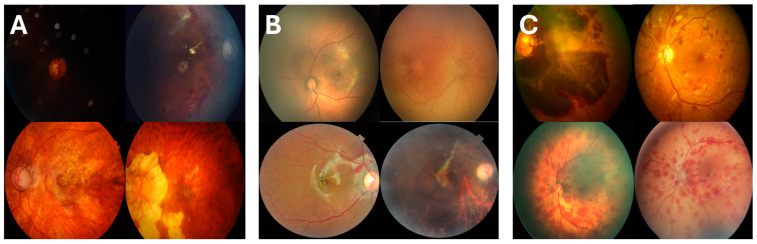
Examples of misclassified cases by the FastViT-SA12 and ResNet18 models: (**A**) Medical cases misclassified as trauma, including poor-focus retinal vein occlusions (**top**) and severe myopia (**bottom**); (**B**) trauma cases misclassified as medical, such as minimal hemorrhages in abusive head trauma (**top**) and subtle choroidal ruptures (**bottom**); (**C**) cases misclassified by ResNet18 but correctly classified by FastViT-SA12, with medical misdiagnoses (**top**) and missed AHT cases (**bottom**). Misclassifications often occurred because of poor image quality, subtle hemorrhages, or complex presentations.

**Figure 3 bioengineering-12-00020-f003:**
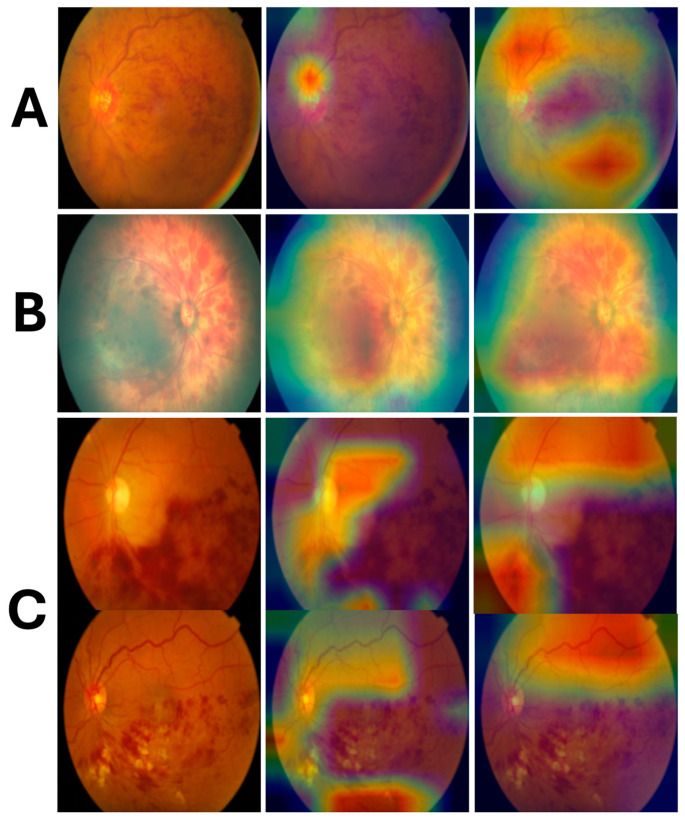
Grad-CAM visualizations for the FastViT-SA12 and ResNet18 models alongside the original fundus images (left: original, middle: FastViT-SA12, right: ResNet18): (**A**) a medical etiology case (vascular occlusion), where FastViT-SA12 highlights the optic disk and focal regions, while ResNet18 emphasizes the arcuate vessels; (**B**) a trauma case (abusive head trauma) with diffuse activations across the fundus; (**C**) retinal vein occlusion cases, where Grad-CAM highlights secondary features like the optic disk or adjacent vessels instead of the hemorrhagic areas.

**Table 2 bioengineering-12-00020-t002:** Performance metrics for the ResNet18 and FastVit_SA12 models.

Class	Positive Predictive Value	Sensitivity	F1-Score
ResNet18 Model (AUC = 0.9626, Accuracy = 94.66%)			
Medical (*n* = 2346)	98.93	94.97	0.9691
Trauma (*n* = 315)	71.15	92.38	0.8039
FastViT_SA12 (AUC = 0.9811, Accuracy = 96.99%)			
Medical (*n* = 2346)	99.35	97.23	0.9828
Trauma (*n* = 315)	82.19	95.24	0.8824

## Data Availability

The data presented in this study are available upon request from the corresponding author. The data are not publicly available due to the sensitive nature of the images, which include fundus photographs with retinal hemorrhages from patients diagnosed with abusive head trauma. All private datasets have been anonymized to protect patient confidentiality. Public datasets, including RFMiD, BRSET, and DeepEyeNet, are accessible through their respective repositories as referenced in the manuscript.

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
