# Peer review of "External Validation of Deep Learning Models for Classifying Etiology of Retinal Hemorrhage Using Diverse Fundus Photography Datasets"

_bioengineering, 2024, doi:10.3390/bioengineering12010020_

Round 1

Reviewer 1 Report

Comments and Suggestions for Authors

The authors raised  a significant clinical need, which is to differentiate  retinal hemorrhage (RH) from etiologies (traumatic vs. medical)—with implications for patient management 

Methology

The methodology is rigorous, using a large data set  (2661) from public and private practice, which corresponds to the real-life management of retinal hemorrhages

The source of your data set : are fundus photograph from USA or form Korea : The multi ethnicity is a strength of the methology, butt other populations are not included in the dataset

Ethic considerations : details data-sharing regulations are needed 

The processing steps, including resizing and normalization, are clearly described

Results

Results are well presented

Performance metrics showed that both models (FastVitSA12and ResNet18) demonstrate strong results, high accuracy and high specificity , indicating potential clinical utility.

Discussion

Strengths and limitations are exposed in the discussion. The authors acknowledge the imbalance in the dataset (fewer trauma cases) and the potential impact on model performance

Add perspective :

-           how physicians can use theses model in clinical implementation ?

-          Futur additional study including  data from diverse populations (European, African, South America ), to improve real-world applicability

Overall, the manuscript presents a valuable contribution to ophthalmological AI. The proposed approach has the potential to improve diagnostic workflows in ophthalmology

Comments on the Quality of English Language

The manuscript is clear , well-written, and pleasant to read

Author Response

Response to Reviewer #1:

We thank the reviewer for their thoughtful and constructive comments, which have helped us strengthen the manuscript.

  1. The authors raised a significant clinical need, which is to differentiate retinal hemorrhage (RH) from etiologies (traumatic vs. medical)—with implications for patient management.

Thank you for recognizing the importance of this clinical need and its implications for improving patient management.

  1. The source of your data set : are fundus photograph from USA or form Korea : The multi ethnicity is a strength of the methodology, but other populations are not included in the dataset

Thank you for highlighting this important aspect. In the revised Discussion, we acknowledge the geographic diversity of the current dataset, which includes fundus images from South Korea and the USA. While this provides a valuable multi-ethnic representation, we recognize the need to further expand the datasets to include underrepresented populations from regions such as Europe, Africa, and South America. Future work will focus on obtaining data from these areas to ensure broader global applicability and improve the generalizability of the model.

Discussion (page 9, paragraph 3):

"Despite the strengths, this study has limitations. The external validation datasets included diverse populations, but underrepresented groups remain, limiting the full generalizability of the models. Expanding datasets to achieve a more global representation is crucial for improving real-world applicability. … By increasing dataset size and incorporating more diverse patient populations, the model could be further refined to better handle challenging or ambiguous cases, thereby enhancing its robustness and diagnostic accuracy. "

  1. Ethic considerations : details data-sharing regulations are needed

We confirm that all private datasets were handled in compliance with the data-sharing regulations of the contributing institutions. The data were anonymized, used exclusively for research purposes, and appropriate agreements were in place. These details have been clarified in the revised Methods section to ensure transparency regarding ethical considerations.

Methods (page 3, section 2.1.1.):

"... Data-sharing and access comply with the respective regulations of the source institutions, and data usage was limited strictly to research purposes as agreed upon with the data providers. …"

  1. Add perspective: How physicians can use these models in clinical implementation?

Thank you for this valuable suggestion. In the revised Discussion, we have added a section outlining the potential clinical applications of these models. Specifically:

  • Automated Screening: The models can assist in triaging patients by quickly identifying cases that require urgent attention, especially in emergency or outpatient settings.
  • Decision Support: By integrating into electronic health record (EHR) systems or diagnostic platforms, the models can provide real-time decision support for ophthalmologists and general practitioners.
  • Telemedicine: The models hold promise for telemedicine applications, enabling remote screening and diagnosis, particularly in underserved or rural areas.
  • Medico-Legal Contexts: The models' ability to differentiate between medical and traumatic retinal hemorrhages may have utility in medico-legal evaluations. However, we emphasize that predictions in such high-stakes scenarios must be used cautiously and always as part of a comprehensive assessment involving clinical judgment and multidisciplinary team input.

            Discussion: (page 9, paragraph 4):

"Integrating these models into real-world clinical workflows has significant potential to enhance diagnostic workflows in ophthalmology. They could assist in triaging patients in emergency and outpatient settings by prioritizing high-risk cases and optimizing resource allocation. In telemedicine settings, they could support remote diagnostics, particularly for underserved populations. Additionally, embedding these models into electronic health record (EHR) systems could enable seamless, automated analysis of fundus images as part of routine workflows. However, while these models demonstrate strong accuracy and specificity, caution is necessary for medico-legal applications. Given the high stakes involved, the models should only be used as adjunctive tools alongside clinical judgment and multidisciplinary team evaluations to mitigate risks associated with potential misclassifications."

  1. Future additional study including data from diverse populations (European, African, South America ), to improve real-world applicability.

            Thank you for your insightful suggestion. In the revised Discussion, we emphasize the importance of expanding the dataset to include populations from underrepresented regions such as Europe, Africa, and South America. This step is crucial for improving the model's real-world applicability and ensuring robust generalization across diverse clinical and demographic settings.

            Discussion (page 9, paragraph 3):

"Despite the strengths, this study has limitations. The external validation datasets included diverse populations, but underrepresented groups remain, limiting the full generalizability of the models. Expanding datasets to achieve a more global representation is crucial for improving real-world applicability. … By increasing dataset size and incorporating more diverse patient populations, the model could be further refined to better handle challenging or ambiguous cases, thereby enhancing its robustness and diagnostic accuracy. "

Final Comment:

We sincerely appreciate the reviewer’s positive evaluation of our work and their valuable suggestions. We believe that the revisions have significantly strengthened the manuscript, enhancing its clarity, clinical relevance, and broader applicability in ophthalmological AI.

Reviewer 2 Report

Comments and Suggestions for Authors

The topic of this paper is interesting. External validation of AI systems plays a crucial role and most of the time is overlooked in research paper. A DL method can work well on internal test set but fail to generalise on external test sets. My comments are listed below:

- significant imbalance between medical (n=2346) and trauma (n=315) cases, this could introduce a bias

- there is a limited description of image quality control measures

- what about model interpretability and/or explainability?

Minor comments::

- there is font inconsistencies in lines 146-149 respect to the rest of the paper

- caption of Table 1 is missing

- Some figures lack clear legends or explanatory text

Author Response

Response to Reviewer #2:

We thank the reviewer for their thoughtful and constructive feedback, and we appreciate the recognition of the importance of our work in addressing the often-overlooked need for external validation in AI systems. Below, we provide detailed responses to each of the comments:

  1. Significant imbalance between medical (n=2346) and trauma (n=315) cases, this could introduce a bias.

Thank you for pointing this out. We acknowledge the imbalance in our dataset and have addressed this limitation in the Discussion section of the revised manuscript. While the model demonstrated robust performance during external validation despite this imbalance, we recognize the importance of improving data representation for trauma cases. Specifically, we aim to collect more trauma-related retinal hemorrhage images, particularly from underrepresented geographic regions and diverse populations, to develop a more balanced dataset. This effort will further enhance the model’s robustness, minimize potential biases, and ensure improved performance in distinguishing between medical and trauma cases.

Methods (page 4, section 3.1.1):

            "… Future studies could include instrumental deliveries to enhance the diversity and gener-alizability of the trauma class. …"

Discussion (page 9, paragraph 3):

"Despite the strengths, this study has limitations. The external validation datasets included diverse populations, but underrepresented groups remain, limiting the full generalizability of the models. Expanding datasets to achieve a more global representation is crucial for improving real-world applicability. … By increasing dataset size and incorporating more diverse patient populations, the model could be further refined to better handle challenging or ambiguous cases, thereby enhancing its robustness and diagnostic accuracy. "

  1. There is a limited description of image quality control measures

            Thank you for raising this point. To ensure image quality consistency, all images underwent standardized preprocessing, including resizing, normalization, and visual inspection to remove low-quality or corrupted images where possible. Additionally, we have included an image quality analysis as part of the interpretability results. As detailed in the Results section, the misclassified cases were often associated with images that had poor focus, low light conditions, or incomplete visibility of key retinal structures (e.g., optic disk or arcuate vessels). These observations highlight the impact of image quality on model performance and the need for incorporating stricter quality control measures in future studies. This clarification has been added to the revised Methods section.

            Methods (page 3, section 2.1.1):

"Image labels were assigned based on the underlying case diagnosis. The determination of hemorrhage presence or absence was entrusted to a panel of ophthalmologists (E.G., R.A., S.Y.K., E.C., D.W.S.). In cases of differing interpretations of an image, consensus was reached through careful discussion and deliberation to arrive at the final classification."

  1. What about model interpretability and/or explainability?

            We appreciate this important comment. In the revised manuscript, we have added a detailed interpretability analysis using Gradient-weighted Class Activation Mapping (Grad-CAM) to explain the decision-making processes of both models. This analysis provides insights into the regions of the fundus images that the models focused on for classification. Key findings include:

  • ResNet18 predominantly highlighted global structural features, such as arcuate vessels.
  • FastViT-SA12 emphasized clinically relevant regions, including the optic disk and focal hemorrhagic areas, which align more closely with diagnostic practices.
  • For medical cases, Grad-CAM activations were focused and localized, whereas trauma cases showed more diffuse, widespread activations across the fundus.
  • In some complex cases, such as retinal vein occlusions, the models focused on adjacent vascular structures or the optic disk instead of the hemorrhagic areas, suggesting reliance on secondary features for classification.

These findings have been incorporated into the Results section and are visualized in Figure 2, providing greater transparency and interpretability of the models’ outputs. This addition highlights the models’ strengths while identifying areas for improvement in handling complex or ambiguous cases.

Results (page 6-7, section 3.2):

"Further analysis of misclassifications (Figure 2) revealed key patterns. FastVit-SA12 missed 8 AHT cases, 1 birth trauma, and 6 choroidal ruptures from the RFMiD test set. Similarly, the ResNet18 model missed 17 AHT cases, 1 birth trauma, and 6 choroidal ruptures. Notably, both models failed to correctly classify all trauma cases from the RFMiD dataset, suggesting the need for greater representation of trauma-related retinal hemorrhages in public datasets to improve model robustness.

The missed cases followed identifiable patterns. Misclassified images were often out of focus or exhibited poor lighting, resulting in incomplete visibility of the optic disk and arcuate vessel structures. Trauma cases with minimal retinal hemorrhages (RH), particularly around the macula, were frequently misdiagnosed as medical. Conversely, medical cases with widespread hemorrhages, such as those caused by retinal vein occlusions or severe myopia, were sometimes classified as trauma."

Results (page 7-8, section 3.3):

"The Grad-CAM visualizations revealed notable differences in the focus areas of the two models. ResNet18 primarily emphasized the arcuate vessels across the fundus (Figure 3A right), which may indicate that the model uses structural vascular patterns to distinguish between medical and trauma-related hemorrhages. In contrast, FastViT-SA12 consistently highlighted the optic disk and specific focal regions within the fundus (Figure 3A middle). This behavior suggests that FastViT-SA12 is more sensitive to localized pathological features, which are often of greater clinical relevance in identifying retinal hemorrhages. The arcuate vessel focus seen in ResNet18 may reflect the model's learning of global structural patterns, while FastViT-SA12's optic disk focus indicates a preference for critical regions commonly evaluated in clinical practice.

Distinct activation patterns emerged when Grad-CAM heatmaps were compared between medical and trauma-related retinal hemorrhage images. In medical cases, the highlighted areas were generally focused and localized, such as the optic disk, arcuate vessels, and specific regions of hemorrhage. Conversely, in trauma-related images, the Grad-CAM outputs demonstrated a diffuse and widespread activation across the fundus (Figure 3B), rather than being limited to specific regions. This broad activation suggests that the models detect subtle, widespread disruptions in the retinal structure, which are characteristic of traumatic hemorrhages.

An unexpected finding emerged in cases of retinal vein occlusion. While prominent hemorrhagic areas were visible in the images, the Grad-CAM heatmaps did not emphasize these regions. Instead, the models highlighted other areas, such as adjacent vascular structures or the optic disk as shown in Figure 3C. This behavior suggests that the models may rely on secondary or global patterns rather than the hemorrhagic areas themselves for classification. These findings highlight the complexity of the decision-making process and the need for further investigation to understand how models prioritize specific features in certain conditions."

  1. Minor Comments (fixed):

- there is font inconsistencies in lines 146-149 respect to the rest of the paper

            Font inconsistencies in lines 146–149 have been corrected.

- caption of Table 1 is missing

            The caption for Table 1 has been added.

- Some figures lack clear legends or explanatory text

            Legends and explanatory text for figures have been updated for improved clarity and readability.

Final Comment:

We sincerely thank the reviewer for their valuable suggestions, which have allowed us to improve the manuscript significantly. By addressing the dataset imbalance, adding image quality control details, and incorporating a thorough interpretability analysis, we believe the revised manuscript provides a more comprehensive and robust evaluation of the models.

Reviewer 3 Report

Comments and Suggestions for Authors

The manuscript, “External Validation of Deep Learning Models for Classifying Etiology of Retinal Hemorrhage Using Diverse Fundus Photography Datasets”, evaluates the performance of two deep learning models, FastVit_SA12 and ResNet18. Both models were independently tested using the same retinal hemorrhage Fundus Photography datasets collected from diverse demographic groups. The study found that the FastVit_SA12 model demonstrated a slight superiority in efficacy compared to ResNet18.

Suggestions to improve the manuscript are given below:

Pg 3, ln 102: In the Data Processing paragraph, it is mentioned that all images were resized to a consistent size of 256 × 256 pixels using bilinear interpolation, which is a standard preprocessing step. However, the sentence in Pg 3 ln 102–103 states: “During testing, images were resized to 224 × 224 pixels and underwent the same per-channel normalization to mirror the preprocessing steps implemented during training.” This inconsistency in image resizing could confuse readers. Providing a justification for selecting 224 × 224 pixels during testing would help clarify the reasoning and maintain consistency in the preprocessing methodology.

Pg 3, ln 128: Retinal hemorrhages resulting from spontaneous vaginal delivery are a relatively common occurrence in newborns, which is generally caused by the pressure exerted on the baby's head during passage through the birth canal. These hemorrhages are typically benign and often resolve on their own without any lasting visual impairment. However, retinal hemorrhage may be more prevalent in deliveries involving instruments such as vacuum extraction. It would therefore be appropriate to clarify why retinal hemorrhages from vaginal deliveries were included in the study, as this could provide valuable context for their relevance to the research.

Pg 4, ln 158: Figure 1 is presented as ‘sample images from different datasets,’ showcasing a montage of retinal hemorrhage fundus photography without captions. To enhance reader understanding, it would be helpful if the authors identified each image with its corresponding diagnostic findings, even if the images are monochrome or multicolor. Additionally, since these images represent the categories (Medical and Trauma etiology) outlined in Table 1, it would be meaningful to link the images to those categories for better contextualization.

Furthermore, cases currently presented in the supplement should be included in the main manuscript, as they add critical value to the study and will help the readers grasp the content better.

In Figure 2, the authors compare the performance of the ResNet18 and FastVit_SA12 models and provide a performance matrix in Table 2. Including a visual comparison of retinal hemorrhage fundus photography images as classified by the two models would allow readers to directly observe and infer the differences between them.

Additionally, the data provided in the supplement (currently in a Word document) should be presented as a comparative figure in the main manuscript. This could be achieved by minimizing the figures and arranging them side by side for a more compact and comprehensive presentation.

Pg 6, ln 186: Figure 3 – It would be helpful to include a more detailed caption below the confusion matrix, as it groups Medical and Trauma cases. Typically, a two-way matrix is used to distinguish between false positives and false negatives. If the authors could provide this data in addition to the current grouping, it would enhance the figure's interpretability and provide greater value to the readers.

Author Response

Response to Reviewer #3:

We thank the reviewer for the valuable feedback and constructive suggestions, which have helped us further clarify and improve the manuscript. Below, we address each comment in detail:

  1. Pg 3, ln 102: In the Data Processing paragraph, it is mentioned that all images were resized to a consistent size of 256 × 256 pixels using bilinear interpolation, which is a standard preprocessing step. However, the sentence in Pg 3 ln 102–103 states: “During testing, images were resized to 224 × 224 pixels and underwent the same per-channel normalization to mirror the preprocessing steps implemented during training.” This inconsistency in image resizing could confuse readers. Providing a justification for selecting 224 × 224 pixels during testing would help clarify the reasoning and maintain consistency in the preprocessing methodology.

            Thank you for pointing out this potential confusion. The two-step resizing process during training was intentional. Images were initially resized to 256 × 256 pixels to ensure uniformity across diverse sources and enable effective data augmentation. Subsequently, the images were resized to 224 × 224 pixels to meet the input size requirements of the transfer learning framework, which is commonly trained on ImageNet at this resolution. During testing, images were resized directly to 224 × 224 pixels to maintain consistency with the model's training pipeline and to reduce computational overhead. This reasoning has been clarified in the Methods section (Page 3, Section 2.2: Data Preprocessing).

 Methods (page 3, section 2.2):
 “To ensure uniformity … to improve model robustness. Following augmentation, images were resized to 224 × 224 pixels to match the input size requirements of the transfer learning framework used in our study. Normalization per channel, based on the ImageNet training set, was applied at all stages. During testing, images were resized directly to 224 × 224 pixels and underwent the same normalization to ensure consistency with the model's input size and training pipeline.”

  1. Pg 3, ln 128: Retinal hemorrhages resulting from spontaneous vaginal delivery are a relatively common occurrence in newborns, which is generally caused by the pressure exerted on the baby's head during passage through the birth canal. These hemorrhages are typically benign and often resolve on their own without any lasting visual impairment. However, retinal hemorrhage may be more prevalent in deliveries involving instruments such as vacuum extraction. It would therefore be appropriate to clarify why retinal hemorrhages from vaginal deliveries were included in the study, as this could provide valuable context for their relevance to the research.

            Thank you for raising this point. We included cases of retinal hemorrhages resulting from spontaneous vaginal delivery as part of the trauma class to represent non-pathological trauma-related etiologies. While these hemorrhages are typically benign and self-limiting, they provide a valuable contrast to more severe trauma-related cases, such as abusive head trauma (AHT). Including these cases allows the model to learn from a broader range of trauma presentations, enhancing its robustness in differentiating trauma-related hemorrhages. We have clarified this rationale in the revised Methods section and addressed the relevance in the Discussion.

Methods (page 4, section 3.1.1):
 “Retinal hemorrhages resulting from spontaneous vaginal delivery were included to represent mild, non-pathological trauma-related cases, providing a broader spectrum of trauma presentations for model training and validation. Future studies could include instrumental deliveries to further enrich the diversity of trauma cases.”

  1. Pg 4, ln 158: Figure 1 is presented as ‘sample images from different datasets,’ showcasing a montage of retinal hemorrhage fundus photography without captions. To enhance reader understanding, it would be helpful if the authors identified each image with its corresponding diagnostic findings, even if the images are monochrome or multicolor. Additionally, since these images represent the categories (Medical and Trauma etiology) outlined in Table 1, it would be meaningful to link the images to those categories for better contextualization.

            Thank you for the suggestion to enhance Figure 1. The figure has been updated to clearly distinguish between medical and trauma etiology images. Each image has been organized into sections, and the captions now specify the diagnostic category represented. This improved presentation links the images directly to the classifications in Table 1, offering better contextualization for readers.

  1. Furthermore, cases currently presented in the supplement should be included in the main manuscript, as they add critical value to the study and will help the readers grasp the content better.
                We appreciate the suggestion to integrate key findings from the supplement into the main text. To address this:

  • Details of misclassified cases have been integrated into the Results section, supported by a new Figure 3 showing examples of misclassifications and patterns observed (e.g., poor image quality, subtle trauma findings).
  • The interpretability analysis using Grad-CAM has been included as part of the main manuscript, with detailed findings presented in Figure 2. This provides greater insights into the decision-making processes of both models.

Results (page 6-7, section 3.2):

"Further analysis of misclassifications (Figure 2) revealed key patterns. FastVit-SA12 missed 8 AHT cases, 1 birth trauma, and 6 choroidal ruptures from the RFMiD test set. Similarly, the ResNet18 model missed 17 AHT cases, 1 birth trauma, and 6 choroidal ruptures. Notably, both models failed to correctly classify all trauma cases from the RFMiD dataset, suggesting the need for greater representation of trauma-related retinal hemorrhages in public datasets to improve model robustness.

The missed cases followed identifiable patterns. Misclassified images were often out of focus or exhibited poor lighting, resulting in incomplete visibility of the optic disk and arcuate vessel structures. Trauma cases with minimal retinal hemorrhages (RH), particularly around the macula, were frequently misdiagnosed as medical. Conversely, medical cases with widespread hemorrhages, such as those caused by retinal vein occlusions or severe myopia, were sometimes classified as trauma."

Results (page 7-8, section 3.3):

"The Grad-CAM visualizations revealed notable differences in the focus areas of the two models. ResNet18 primarily emphasized the arcuate vessels across the fundus (Figure 3A right), which may indicate that the model uses structural vascular patterns to distinguish between medical and trauma-related hemorrhages. In contrast, FastViT-SA12 consistently highlighted the optic disk and specific focal regions within the fundus (Figure 3A middle). This behavior suggests that FastViT-SA12 is more sensitive to localized pathological features, which are often of greater clinical relevance in identifying retinal hemorrhages. The arcuate vessel focus seen in ResNet18 may reflect the model's learning of global structural patterns, while FastViT-SA12's optic disk focus indicates a preference for critical regions commonly evaluated in clinical practice.

Distinct activation patterns emerged when Grad-CAM heatmaps were compared between medical and trauma-related retinal hemorrhage images. In medical cases, the highlighted areas were generally focused and localized, such as the optic disk, arcuate vessels, and specific regions of hemorrhage. Conversely, in trauma-related images, the Grad-CAM outputs demonstrated a diffuse and widespread activation across the fundus (Figure 3B), rather than being limited to specific regions. This broad activation suggests that the models detect subtle, widespread disruptions in the retinal structure, which are characteristic of traumatic hemorrhages.

An unexpected finding emerged in cases of retinal vein occlusion. While prominent hemorrhagic areas were visible in the images, the Grad-CAM heatmaps did not emphasize these regions. Instead, the models highlighted other areas, such as adjacent vascular structures or the optic disk as shown in Figure 3C. This behavior suggests that the models may rely on secondary or global patterns rather than the hemorrhagic areas themselves for classification. These findings highlight the complexity of the decision-making process and the need for further investigation to understand how models prioritize specific features in certain conditions."

  1. In Figure 2, the authors compare the performance of the ResNet18 and FastVit_SA12 models and provide a performance matrix in Table 2. Including a visual comparison of retinal hemorrhage fundus photography images as classified by the two models would allow readers to directly observe and infer the differences between them.
                Thank you for the suggestion. While the performance matrices for both models are summarized in Table 2, we have relocated the confusion matrix figure (previous Figure 2) to the supplementary materials to streamline the main manuscript. Additionally, we have added new figures (Figure 2 and Figure 3) to visualize key findings, including Grad-CAM activations and misclassification patterns. These new visualizations provide a clear comparison of the models’ outputs and highlight areas for improvement.

  2. Additionally, the data provided in the supplement (currently in a Word document) should be presented as a comparative figure in the main manuscript. This could be achieved by minimizing the figures and arranging them side by side for a more compact and comprehensive presentation.
                Thank you for this feedback. We have added a more detailed caption to the confusion matrix figure (now in the supplementary materials), specifying the breakdown of false positives and false negatives for medical and trauma classifications. This enhances interpretability and provides greater value to readers.

                Additionally, the updated figures for misclassified cases and interpretability analysis offer complementary insights into the models’ performance, addressing the reviewer’s request for visual comparisons.

Final Comment:
            We sincerely thank the reviewer for these thoughtful and actionable suggestions. By addressing the concerns related to image resizing, dataset inclusion, figure clarity, and supplementary content integration, we believe the manuscript has been significantly improved in clarity, comprehensiveness, and presentation. The revised figures and captions now provide a clearer understanding of the study findings, and the discussion offers a deeper exploration of model strengths, limitations, and future directions.

Round 2

Reviewer 2 Report

Comments and Suggestions for Authors

The manuscript improved after revision. The authors addressed my main comments. 

Reviewer 3 Report

Comments and Suggestions for Authors

The manuscript is considerably revised and is presented adequately.